# Online search interest in long-term symptoms of coronavirus disease 2019 during the COVID-19 pandemic in Japan: Infodemiology study using the most visited search engine in Japan

Kosuke Ishizuka[1]*, Taiju Miyagami[2], Tomoya Tsuchida[3]☯, Mizue Saita[2]☯, Yoshiyuki Ohira[3]☯, Toshio Naito[2]☯

1 Department of General Medicine, Yokohama City University School of Medicine, Yokohama, Japan,
2 Department of General Medicine, Juntendo University Faculty of Medicine, Tokyo, Japan, 3 Division of General Internal Medicine, Department of Internal Medicine, St. Marianna University School of Medicine, Kawasaki, Kanagawa, Japan

☯ These authors contributed equally to this work.
* e103007c@yokohama-cu.ac.jp

**Data Availability Statement:** All relevant data are within the manuscript and its Supporting Information files.

## Abstract

Prolonged symptoms that occur after COVID-19 (long-COVID) vary from mild, which do not interfere with daily life, to severe, which require long-term social support. This study assessed the secular trend in online searches on long-COVID in Japan. We conducted an observational study using data provided by Yahoo! JAPAN on the monthly search volume of query terms related to long-COVID from January 2020 to December 2022, including the search volume of the query "コロナ後遺症" (long-COVID in Japanese). The number of new cases of COVID-19 by month was used as a control for search trends, and the symptoms retrieved in conjunction with long-COVID were compared. Trends in online searches for each symptom of long-COVID were analyzed. The symptoms of long-COVID were classified according to "Component 1—Symptoms and Complaints" of the International Classification of Primary Care, 2nd edition (ICPC-2). Interest in long-COVID increased in response to peaks in the number of new cases of COVID-19 in Japan. The most frequent symptom searches with long-COVID were hair loss/baldness (3,530, 21,400, and 33,600 searches in 2020, 2021, and 2022, respectively), cough (340, 7,900 and 138,910 searches in 2020, 2021, and 2022, respectively), disturbance of smell/taste (230, 13,340, and 44,160 searches in 2020, 2021, and 2022, respectively), and headache (580, 6,180, and 42,870 searches in 2020, 2021, and 2022, respectively). In addition, the ranking of interest in "weakness/tiredness, general" in long-COVID increased each year (not in the top 10 in 2020, seventh in 2021, and second in 2022), and the absolute number of searches also increased. To our knowledge, this is the first study to investigate secular trends in online interest in long-COVID in the world. Continued monitoring of online interest in long-COVID is necessary to prepare for a possible increase in the number of patients with long-COVID.

**Funding:** The authors received no specific funding for this work.

**Competing interests:** The authors have declared that no competing interests exist.

## Introduction

The coronavirus disease (COVID-19) pandemic started with an outbreak of pneumonia of unknown cause in Wuhan, China in December 2019, and subsequently spread worldwide including Japan [1]. Extensive information has been accumulated on the acute phase symptoms of COVID-19 have been accumulated, and countermeasures against infection and methods have been established for diagnosis, treatment, and prevention. However, some individuals develop various prolonged symptoms after severe acute respiratory syndrome coronavirus 2 (SARS-CoV-2) infection (long-COVID) [2]. The term "long-COVID" first entered public discourse on May 20, 2020, when Dr. Elisa Perego shared #longcovid on Twitter, now known as X [3–5]. A hashtag (#) is a concept-labelling tool that encourages public sharing and worldwide dissemination of the discussed concept by grouping conversations around it. Initially, #longcovid referred to Perego's experience of prolonged "cyclical, multiphasic, and multisystem" symptoms of COVID-19 [5]. Twitter, with approximately 187 million users worldwide at the time, emerged as a platform for individuals with persistent COVID-19 symptoms to publicly share their experiences, symptoms, and concerns associated with living with the condition [5, 6]. Subsequently, on October 6, 2021, the World Health Organization (WHO) defined long-COVID as the onset of symptoms within 3 months after the onset of COVID-19 and persistence of symptoms for more than 2 months, excluding other diseases [7, 8]. In a large matched cohort study conducted in the Netherlands, long-COVID was reported in 12.7% of patients after COVID-19 [9]. Long-COVID symptoms vary from mild, which do not interfere with daily life, to severe, which require long-term social support [2].

Long-COVID has been extensively studied, and symptoms vary widely and include fatigue, dyspnea, olfactory disturbance, taste disorder, headache, chest pain, memory disturbance, and hair loss [2, 9–13]. However, the study settings and follow-up methods differed from survey to survey; therefore, the results should be interpreted with caution.

The internet is a common source of reference for information on disease and health, and internet usage influences decisions regarding initiation of care and choice of treatment in people living with long-COVID. Infodemiology (i.e., information epidemiology) is a field in health informatics defined as "the science of distribution and determinants of information in an electronic medium, specifically the internet, or in a population, with the ultimate aim to inform public health and public policy" [14]. During the COVID-19 pandemic, infodemiology studies found that online search trends for symptoms associated with COVID-19 coincided with the course of the pandemic [15]. Although face-to-face surveys of outpatients and large web surveys have been conducted on symptoms related to long-COVID, there is a lack of data on the secular trends of internet searches using search engines. The purpose of this study was to characterize the secular trend in online searches on long-COVID using search engines in Japan.

## Methods

### Data sources

In order to investigate the patterns of online searches for long-COVID during the COVID-19 pandemic, we used the search volume for a certain period extracted from Yahoo! JAPAN, which is one of the most frequently used search engines in Japan. Yahoo! JAPAN is the most accessed site in Japan, and compared with other countries, the utilization rate of Google in Japan is low. According to the data in Japan, yahoo.co.jp was accessed 1.18 billion times in November 2021, and the Japanese version of Google was accessed 617 million times in the same month [16]. The search volume data were obtained from the server of Yahoo! Japan DS.

INSIGHT (last accessed on June 3, 2023). Yahoo! Japan DS. INSIGHT can be accessed by people working at educational and research institutions by obtaining a login ID. It is a tool that enables analysis of all big behavioral data of Yahoo! Japan, such as keywords, time period, gender, age, and prefecture, and the transitions and trends in these searches over time (S1 Table). The user manual can be found at https://datasolution.yahoo-net.jp/view/knowledgebase/641.

The internet user population was defined as the number of internet users throughout Japan, calculated based on the "Telecommunications Usage Trends Survey," published by the Japan Ministry of Internal Affairs and Communications.

The number of new COVID-19 cases was obtained from data published by the Japan Ministry of Health, Labour and Welfare [17].

## Search queries used in the analysis

In order to investigate the trend of online searches on long-COVID, we conducted an observational study using the monthly search volume of query terms obtained from Yahoo! JAPAN. The search term "コロナ後遺症" (long-COVID in Japanese) was used to measure the online level of general interest in long-COVID from January 2020 to December 2022, and the results were presented with the number of new cases of COVID-19 reported in Japan. We also compared the symptoms searched with long-COVID and analyzed the difference in online interest in each long-COVID symptom from January 2020 to December 2022, by year. Long-COVID symptoms were classified according to "Component 1—Symptoms and complaints" of the International Classification of Primary Care, 2nd edition (ICPC-2) [18]. The ICPC focuses on an analysis of individuals' "reason for the encounter," and it has the advantage that it can directly describe the reason for the encounter from a patient perspective as well as from a healthcare provider perspective. In addition, we obtained baseline data on searches for each of these symptoms during the pandemic as distinct searches, not paired with long-COVID. In this study, we conducted infodemiology using data on internet searches using the Yahoo search engine. Two researchers (KI and TM) conducted independent evaluations of the Yahoo search engine data for symptom classification in order to minimize observer bias.

## Data standardization

Data were obtained on the monthly number of searches per prefecture for each search query, adjusted by gender, and converted to standardized z scores using the following formula:

$$z\ score_i = \frac{Search\ volume\ of\ query\ A_i - \overline{Search\ volume\ of\ query\ A}}{Standard\ deviation\ of\ search\ volume\ of\ query\ A}$$

$$\overline{Search\ volume\ of\ query\ A} = \frac{\sum_{i=1}^{n} Search\ volume\ of\ query\ A}{n}$$

$$Standard\ deviation\ of\ search\ volume\ of\ query\ A$$

$$= \sqrt{\frac{\sum_{i=1}^{n} \left(Search\ volume\ of\ query\ A_i - \overline{Search\ volume\ of\ query\ A}\right)^2}{n-1}}$$

"query A" refers to the queried search term.

## Statistical analysis

Internet users were stratified according to their age on the day of the search, gender, and search year. Demographics were summarized as the number and percentage of users for

**Table 1. Number of searches for long-COVID using the Yahoo! JAPAN search engine, by gender and age in the years 2020, 2021, and 2022.**

| | | Year 2020 | | | | Year 2021 | | | | Year 2022 | | | |
|---|---|---|---|---|---|---|---|---|---|---|---|---|---|
| | | Internet user [a] | | Search term "Long-COVID" | | Internet user [a] | | Search term "Long-COVID" | | Internet user [a] | | Search term "Long-COVID" | |
| | | Search volume | (%) | Search volume | (%) | Search volume | (%) | Search volume | (%) | Search volume | (%) | Search volume | (%) |
| Overall | | | | | | | | | | | | | |
| | | 96,688,079 | (100%) | 75,900 | (100%) | 99,499,669 | (100%) | 164,800 | (100%) | 99,499,669 | (100%) | 594,000 | (100%) |
| Gender | | | | | | | | | | | | | |
| | Female | 47,568,804 | (49%) | 46,900 | (62%) | 49,212,761 | (49%) | 103,000 | (62%) | 49,212,761 | (49%) | 409,000 | (69%) |
| | Male | 49,119,275 | (51%) | 29,000 | (38%) | 50,286,908 | (51%) | 61,800 | (38%) | 50,286,908 | (51%) | 185,000 | (31%) |
| Age group (year) | | | | | | | | | | | | | |
| | <20 | 13,386,518 | (14%) | 4,100 | (5%) | 13,787,372 | (14%) | 7,800 | (5%) | 13,787,372 | (14%) | 26,300 | (4%) |
| | 20–29 | 11,777,729 | (12%) | 6,300 | (8%) | 12,432,946 | (13%) | 13,200 | (8%) | 12,432,946 | (13%) | 40,600 | (7%) |
| | 30–39 | 13,547,255 | (14%) | 12,200 | (16%) | 13,615,333 | (14%) | 24,400 | (15%) | 13,615,333 | (14%) | 91,500 | (15%) |
| | 40–49 | 17,429,796 | (18%) | 22,700 | (31%) | 17,491,018 | (17%) | 48,100 | (29%) | 17,491,018 | (17%) | 181,000 | (31%) |
| | 50–59 | 15,451,255 | (16%) | 17,800 | (23%) | 16,258,923 | (16%) | 42,900 | (26%) | 16,258,923 | (16%) | 152,000 | (26%) |
| | 60–69 | 12,716,666 | (13%) | 8,900 | (12%) | 12,887,755 | (13%) | 19,500 | (12%) | 12,887,755 | (13%) | 73,000 | (12%) |
| | ≥70 | 12,378,860 | (13%) | 3,900 | (5%) | 13,026,322 | (13%) | 8,700 | (5%) | 13,026,322 | (13%) | 29,000 | (5%) |

[a] The internet user population is the number of internet users throughout Japan, calculated based on the "Telecommunications Usage Trends Survey" published by the Ministry of Internal Affairs and Communications.

コロナ後遺症: Long COVID in Japanese.

Source: Yahoo Japan Data Solution DS. INSIGHT

categorical variables. Descriptive statistics were calculated using Microsoft Excel (Microsoft Corporation, Redmond, WA, USA).

## Ethics

This study adhered to the ethical guidelines for Medical and Health Research Involving Human Subjects of the Ministry of Health, Labour and Welfare of Japan [19]. In accordance with this guideline, since this study used previously anonymized and de-identified data, an ethical review was waived, and patient informed consent was not required. Additionally, it conformed to the ethical principles set forth in the Declaration of Helsinki (Fortaleza Revision, 2013) [20].

## Results

Between 2020 and 2022, 50–51% of all adult internet users were male, and the user population was relatively evenly distributed across different age groups (Table 1).

Online interest in long-COVID increased in mid-2021, early 2022, and late 2022, corresponding to peaks in the number of new cases of COVID-19 reported during waves of the COVID-19 epidemic in Japan (Fig 1). Although the gender distribution varied from 2020 to 2022, those aged 40–49 years consistently showed the highest online interest in long-COVID (29–31%), followed by those aged 50–59 years (23–26%) and 30–39 years (15–16%) (Table 1). Additionally, the data on the number of annual searches per capita by prefecture are shown in S2 Table.

Classification of the symptoms searched with long-COVID into ICPC-2 terms by two independent researchers (KI and TM) was in complete agreement. The search volume for long-COVID in 2020 was 75,900 searches, and the most frequent symptom combined with long-COVID searches was "hair loss/baldness" (3,530 searches), followed by "headache" (580 searches), "throat symptom/complaint" (520 searches), "fever" (490 searches), and "cough"

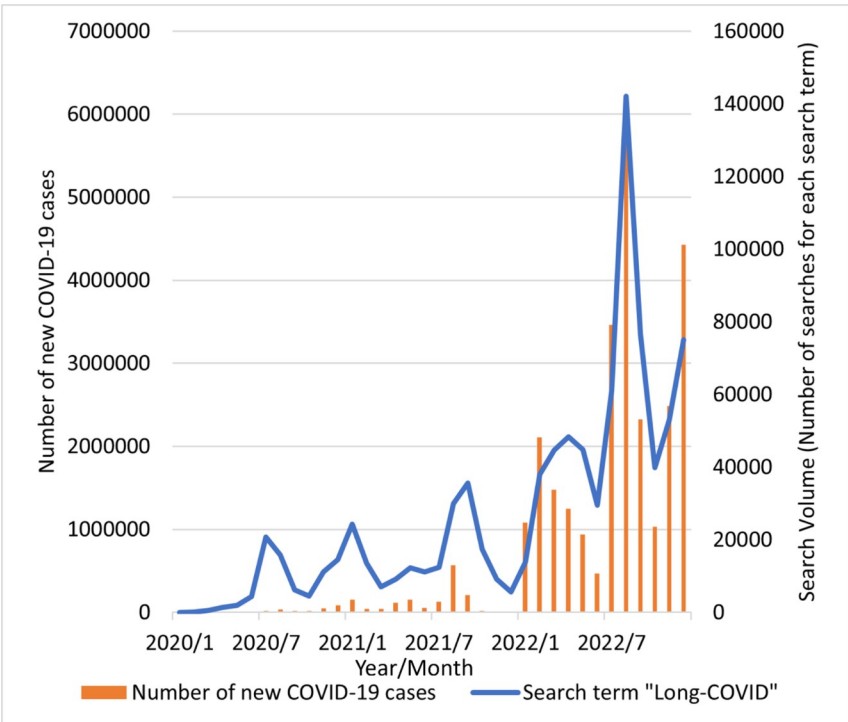

**Fig 1. Monthly online long-COVID search volume and number of new COVID-19 cases reported in Japan from January 2020 to December 2022.** Online interest in long-COVID increased in mid-2021, early 2022, and late 2022, corresponding to peaks in the number of new cases of COVID-19 reported during waves of the COVID-19 epidemic in Japan. Source: Yahoo Japan Data Solution DS. INSIGHT.

(340 searches) followed (Table 2). The search volume for long-COVID increased to 164,800 searches in 2021, and the most frequent symptom combined with long-COVID searches was "hair loss/baldness" (21,400 searches), as in 2020, followed by "disturbance of smell/taste" (13,340 searches), "cough" (7,900 searches), "fever" (7,290 searches), and "headache" (6,180 searches) (Table 2). The search volume for long-COVID further increased to 594,000 searches in 2022, and the most frequent symptom combined with long-COVID searches was "cough" (138,910 searches), followed by "weakness/tiredness, general" (52,270 searches), "disturbance of smell/taste" (44,160 searches), "headache" (42,870 searches), and "hair loss/baldness" (33,600 searches) (Table 2). In addition, Table 3 shows the search data for each symptom included in Table 2 as distinct searches, not paired with long-COVID.

## Discussion

In this study, the trend in online searches for long-COVID in Japan increased during pandemic waves of COVID-19 according to number of new cases of COVID-19 reported. By definition, long-COVID symptoms appear months after COVID-19. Therefore, although internet users were worried about long-COVID, these findings may represent persistent acute symptoms present before the time threshold for long-COVID onset. The highest ranked symptoms searched with long-COVID were hair loss/baldness, cough, disturbance of smell/taste, and headache. General weakness/tiredness is reported to be the most frequent symptom of long-COVID [21], and the online search ranking of "weakness/tiredness, general" increased each year (not ranked in the top 10 in 2020, seventh in 2021, and second in 2022), and the absolute

**Table 2. Long-COVID online search volume and top 10 symptoms (2020–2022).**

| Year | 2020 | 2021 | 2022 |
|---|---|---|---|
| Number of searches for long-COVID (searches) | Number of searches for long-COVID (75,900) | Number of searches for long-COVID (164,800) | Number of searches for long-COVID (594,000) |
| No. 1 (searches) | Hair loss/Baldness (3,530) | Hair loss/Baldness (21,400) | Cough (138,910) |
| No. 2 (searches) | Headache (580) | Disturbance of smell/taste (13,340) | Weakness/Tiredness, general (52,270) |
| No. 3 (searches) | Throat symptom/complaint (520) | Cough (7,900) | Disturbance of smell/taste (44,160) |
| No. 4 (searches) | Fever (490) | Fever (7,290) | Headache (42,870) |
| No. 5 (searches) | Cough (340) | Headache (6,180) | Hair loss/Baldness (33,600) |
| No. 6 (searches) | Disturbance of smell/taste (230) | Shortness of breath/Dyspnea (4,940) | Throat symptom/complaint (32,550) |
| No. 7 (searches) | Joint symptom/complaint NOS (180) | Weakness/Tiredness, general (4,400) | Fever (25,710) |
| No. 8 (searches) | Chest symptom/complaint (180) | Memory disturbance (4,220) | Vertigo/Dizziness (24,050) |
| No. 9 (searches) | Vertigo/dizziness (130) | Feeling depressed (2,890) | Shortness of breath/Dyspnea (20,100) |
| No. 10 (searches) | Shortness of breath/Dyspnea (120) | Muscle pain (2,830) | Feeling depressed (16,900) |

Source: Yahoo Japan Data Solution DS. INSIGHT

number of online searches also increased. Furthermore, Table 3 clarified that a steady baseline during the pandemic for these isolated search terms not paired with long-COVID.

Hair loss/baldness was the first ranked symptom combined with long-COVID in 2020 and 2021, and the fifth ranked symptom in 2022. In a survey of patients recovering from COVID-19 in Japan, alopecia was reported to occur in 24.1% of cases [22]. Alopecia associated with long-COVID is generally resting alopecia, a condition in which hair falls out due to the rapid transition of growing hair into the resting phase [22]. Many symptoms that occur after COVID-19 tend to improve within approximately 50 days after the acute phase, but alopecia tends to develop approximately 2 months after the onset of the disease and recovers after approximately 3 months [22, 23]. Therefore, although the patients had symptoms other than alopecia during the acute phase that were highly disruptive to their daily lives, these symptoms improved, whereas alopecia persisted, suggesting that there may have been a high level of interest in long-COVID-related alopecia online from an aesthetic point of view.

Among the symptoms searched with long-COVID in the online interest, cough occupied the fifth place in 2020, the third place in 2021, and the first place in 2022. This suggests that the frequency of cough in long-COVID may be increasing compared to other symptoms in comparison with that before the emergence of the SARS-CoV-2 Delta variant. In addition, the frequency of post-infectious cough is higher in cases of cough of less than eight weeks. There is no specific treatment for post-infectious cough, although it gradually improves with symptomatic treatment. It is considered that when post-infectious cough occurs after COVID-19, even after a period of recuperation associated with COVID-19, patients may think that their cough may be considered infectious by others because of the persistence of the cough, or patients may be concerned about what others think of them, and thus cough is often searched as a post-corona sequela in search engines even if symptoms have not persisted for more than two months, as defined in the long-COVID.

**Table 3. Online search volume for the top 10 symptoms included in Table 2 as distinct searches, not paired with long-COVID (2020–2022).**

| Symptom | 2020 | 2021 | 2022 |
|---|---|---|---|
| | Number of searches | Number of searches | Number of searches |
| Hair loss/Baldness | 237,200 | 231,630 | 204,980 |
| Headache | 450,700 | 515,000 | 462,900 |
| Throat symptom/complaint | 398,060 | 327,570 | 493,720 |
| Fever | 201,900 | 169,500 | 194,200 |
| Cough | 67,800 | 43,800 | 51,400 |
| Disturbance of smell/taste | 95,400 | 67,200 | 54,800 |
| Joint symptom/complaint NOS | 46,840 | 46,140 | 38,090 |
| Chest symptom/complaint | 328,000 | 290,000 | 242,000 |
| Vertigo/Dizziness | 127,980 | 144,680 | 140,500 |
| Shortness of breath/Dyspnea | 82,460 | 82,020 | 59,080 |
| Weakness/Tiredness, general | 132,630 | 132,240 | 122,260 |
| Memory disturbance | 42,340 | 62,190 | 51,110 |
| Feeling depressed | 75,400 | 87,900 | 72,300 |
| Muscle pain | 85,000 | 82,600 | 64,900 |

Source: Yahoo Japan Data Solution DS. INSIGHT

Disturbance of smell/taste ranked sixth, second, and third in 2020, 2021, and 2022, respectively, in the list of symptoms searched for in combination with long-COVID. In cases of long-COVID with long-lasting damage, it is suggested that olfactory nerve cells are damaged as a pathological condition [24]. The incidence of disturbance of smell/taste has been suggested to be affected by mutations in the mutant strains [25, 26], and this may also be the case in long-COVID.

Headache was the second most common symptom searched with long-COVID in 2020, fifth in 2021, and fourth in 2022. In a meta-analysis, headache was reported to be present in 19.8% of patients 60 days after COVID-19 onset [27]. Although acute headache can be caused by aggravation of existing headache due to COVID-19, and new migraine-like headache can be caused by activation of trigeminal vasculature by COVID-19, persistent headache in long-COVID occurs late and is not necessarily preceded by headache in the acute stage of COVID-19 [28, 29]. Pathophysiological factors such as changes in glutamate levels and hypoxic damage may be involved in headache in long-COVID [29]. The prevalence of headache is high in long-COVID and various pathological conditions have been implicated. This study suggests that online interest in headache in long-COVID has remained high over time.

General weakness/tiredness is reported to occur in 23–42% of patients with long-COVID and is the most frequent symptom reported among COVID-19 sequelae [21]. In this study, general weakness/tiredness was not ranked within the top ten symptoms searched with long-COVID in 2020 but ranked seventh in 2021 and second in 2022. It is possible that general weakness/tiredness in long-COVID increased more than other symptoms after the epidemic wave caused by the Omicron variant compared with symptoms experienced before the Delta wave. Postural orthostatic tachycardia syndrome (POTS) and myalgic encephalomyelitis/chronic fatigue syndrome (ME/CFS) are the most frequent causes of general weakness/tiredness in long-COVID [30–32], and it is important to treat the symptom according to the cause. Psychiatric symptoms are reported to occur in 62% of patients after COVID-19 [33] and are a major cause of general weakness/tiredness in long-COVID. In this study, feeling depressed ranked ninth in 2021 and tenth in 2022 in the online searches combined with long-COVID. In

patients with long-COVID, a comprehensive approach is necessary, not only for physical assessment, but also for psychological care and social reintegration support by social workers [13, 33–35]. The increase in online interest in general weakness/tiredness in long-COVID after the Omicron wave suggests that the need for a comprehensive approach to general weakness/tiredness in long-COVID may have increased over time.

This study showed a growing awareness and interest in long-COVID and documents the secular trends in the number of online searches related to long-COVID symptoms during the COVID-19 pandemic. To our knowledge, this is the first study to investigate secular trends in online interest in long-COVID. It is important to monitor interest in long-COVID on an ongoing basis to fully understand its epidemiology and the medical needs of individuals experiencing long-COVID symptoms. Additionally, a comprehensive approach is necessary for the support of individuals with long-COVID, not only for physical assessment, but also for psychological care and social reintegration support by social workers [13, 33–35]. Although this study focused on online interest in long-COVID symptoms, it studies on online interest in long-COVID treatment would also be informative.

## Limitations

This study has several limitations. First, the data source was limited to Yahoo! JAPAN and does not include internet searches conducted using other search engines. This limits the generalizability of the results. Second, data obtained from online search engines may be subject to non-representative sampling and methodological biases specific to the search platform. However, in this study, we tried to prevent these biases by an independent evaluation of Yahoo search engine data by two researchers. Third, an analysis of composite search volumes consisting of multiple search terms or further narrowed search terms may further increase the significance of these results. Fourth, the search volume may increase with a longer duration of symptoms. Fifth, the number of internet users and the number of users of the Yahoo! JAPAN search engine did not increase between 2020 and 2022, which may not be related to the trend in Japan. Sixth, internet users may search for long-COVID symptoms when prolonged symptoms occur after COVID-19, even if the symptoms do not persist for more than 2 months, which is the definition of long-COVID. Seventh, stratification by age and gender represents the Yahoo! user login info but may not reflect the current user because Yahoo! IDs can be used by multiple users with different demographics. For example, users in a family may access Yahoo! using the Yahoo! ID of another family member. Eighth, the linked dataset is written in Japanese, not English. We have added an explanation of Yahoo! Japan DS. INSIGHT functions in English in the main text and S1 Table for readers who cannot read Japanese. Ninth, because search data was only available after 2020, when the pandemic began, we could not determine whether the number of searches represent an absolute increase in the number of searches for these symptoms or simply fluctuations in the number of searches over time during the period when COVID-19 was prevalent.

## Conclusion

In conclusion, this study showed the growing awareness and interest in long-COVID and the secular trends of online search engines on long-COVID during the COVID-19 pandemic. This is the first study to investigate secular trends in online interest in long-COVID in the world. The number of online searches for long-COVID was linked to the number of confirmed cases of COVID-19, and hair loss/baldness, cough, disturbance of smell/taste, and headache were the most frequently searched symptoms. In addition, the number of searches for general tiredness/weakness increased over time, so continued monitoring is necessary to

prepare for the possible increase in the number of patients with long-COVID who will be seen in the future.

## Supporting information

**S1 Table. The explanation of Yahoo! Japan DS. INSIGHT functions.**
(PDF)

**S2 Table. Annual number of searches for long-COVID by prefecture.**
(PDF)

## Acknowledgments

The authors thank the physicians who participated in the present study.

## Author Contributions

**Conceptualization:** Kosuke Ishizuka, Taiju Miyagami, Tomoya Tsuchida, Mizue Saita, Yoshiyuki Ohira, Toshio Naito.

**Data curation:** Kosuke Ishizuka, Taiju Miyagami.

**Formal analysis:** Kosuke Ishizuka, Taiju Miyagami.

**Investigation:** Kosuke Ishizuka, Taiju Miyagami.

**Methodology:** Kosuke Ishizuka, Taiju Miyagami.

**Project administration:** Toshio Naito.

**Resources:** Kosuke Ishizuka, Taiju Miyagami.

**Software:** Kosuke Ishizuka, Taiju Miyagami.

**Supervision:** Taiju Miyagami.

**Validation:** Taiju Miyagami, Tomoya Tsuchida, Mizue Saita, Yoshiyuki Ohira, Toshio Naito.

**Visualization:** Kosuke Ishizuka, Taiju Miyagami.

**Writing – original draft:** Kosuke Ishizuka.

**Writing – review & editing:** Taiju Miyagami, Tomoya Tsuchida, Mizue Saita, Yoshiyuki Ohira, Toshio Naito.

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
