## [Decision Letter · Decision Letter 0]

22 Sep 2023

PONE-D-23-28136Online Search Interest in long-term symptoms of coronavirus disease 2019 During the COVID-19 Pandemic in Japan: Infodemiology Study Using the Most Visited Search Engine in JapanPLOS ONE

Dear Dr. Ishizuka,

Thank you for submitting your manuscript to PLOS ONE. After careful consideration, we feel that it has merit but does not fully meet PLOS ONE’s publication criteria as it currently stands. Therefore, we invite you to submit a revised version of the manuscript that addresses the points raised during the review process.

ACADEMIC EDITOR: 

The manuscript needs major revisions, please respond step by step to the reviewers.

Best regards

We look forward to receiving your revised manuscript.

Kind regards,

Omar Enzo Santangelo

Academic Editor

PLOS ONE

Journal Requirements:

Reviewers' comments:

Reviewer's Responses to Questions

**Comments to the Author**

1. Is the manuscript technically sound, and do the data support the conclusions?

Reviewer #1: Partly

Reviewer #2: Yes

2. Has the statistical analysis been performed appropriately and rigorously? 

Reviewer #1: Yes

Reviewer #2: I Don't Know

3. Have the authors made all data underlying the findings in their manuscript fully available?

Reviewer #1: No

Reviewer #2: No

4. Is the manuscript presented in an intelligible fashion and written in standard English?

Reviewer #1: Yes

Reviewer #2: Yes

5. Review Comments to the Author

Reviewer #1: This is a paper on Online Search Interest in long-term symptoms of coronavirus disease 2019 During the COVID-19 Pandemic in Japan: Infodemiology Study Using the Most Visited Search Engine in Japan. I have some major and minor comments on your interesting and important paper. The comments are listed below.

Major comments:

1. Abstract: The symptoms are listed what you found in this study. I suggest that you add more numerical/statistical information on your results. You have all this information available in tables.

2. Introduction: Only two paragraphs introduce the topic, the first introduces long-COVID, and second briefly says that “there is a lack of data on the secular trends of internet searches using search engines.” More information (literature published) on internet searching in general is needed to add in Introduction, since this section is now too short and should be much longer in order to introduce the field of the research topic.

3. Introduction: Related to my previous comment, I suggest that you consider using a hypothesis based on the relation of long-COVID and internet searching.

4. More information is needed to show where your study results can be applied to. Also, some information on future work/studies should be added in order to show which interest areas of online information seeking and/or long-COVID research should be focused on in the future.

Minor comments:

5. The term ”infodemiology” is used in this paper. Please briefly explain in text what this means.

6. “Internet” and “internet” are used in text, sometimes with a capital letter, sometimes without. Please choose one of them and use consistently throughout.

7. References: Please use PMIDs or DOIs at the end of each scientific reference.

Reviewer #2: The authors perform an internet search analysis related to Long COVID and symptoms. They use Yahoo analytics, analyze the dataset retrieved, and interpret the findings.

Questions/comments:

1. Table 2: commas are misplaced in the numbers for the top row.

2. Page 11, line 90: Link to the manual doesn't work.

3. Data accessibility: The dataset should be provided directly, or clearer process for how readers might directly obtain the data should be provided. The links to the Japanese language websites were unclear as to how the data might be downloaded for independent analysis.

4. Comparison/adjudication of the evaluations performed by the independent researchers to minimize bias does not appear to be reported. What was the process followed and how often were there differences?

5. Data by prefecture is noted as retrieved yet not reported. This may be of interest to readers including per capita annual searches by prefecture and implications.

6. Stratification by age and sex are reported yet the reliability of this information is not discussed. How does Yahoo! have access to this data? If this merely represents the Yahoo! user login info, this may be inaccurate, not reflect the current user, and, aren't many searches anonymous? For example, is there published data for how often other users in a family access Yahoo! that's already logged in under another family member? The method by which user age/sex are inferred and these limitations should be discussed fully along with any references that validate usage of these data elements in this manner.

7. Did the authors consider obtaining baseline search data for each of these symptoms pre-pandemic and also during the pandemic as distinct searches, not paired with Long Covid? This would clarify whether these symptom findings represent an absolute increase in searches for these symptoms or simply shifted time-wise when COVID was prevalent. In addition, the isolated symptom trend if including prior years would clarify whether a steady baseline existed pre-pandemic for these isolated search terms or whether, for some reason, there was already a positive total trend that's just clumping around COVID peaks.

8. Since Long COVID symptoms by definition would appear months after COVID infection, the authors should address why peaks in Long COVID searches appear to coincide almost exactly with the number of active cases and are not delayed.

Overall - nice analysis. Including baselines for the symptoms analyzed would make the paper stronger and more compellingly linked to COVID than search epiphenomena clustered during outbreaks. Absence of a time shift in the search also introduces the notion that these findings represented acute symptoms persisting prior to the the Long COVID time threshold although patients were worried about long COVID. One other point that may be of interest would be putative COVID (and maybe Long COVID) treatments paired to these search terms and trends over time.

6. PLOS authors have the option to publish the peer review history of their article (what does this mean?). If published, this will include your full peer review and any attached files.

Reviewer #1: No

Reviewer #2: **Yes: **David Liebovitz, MD

---

## [Author Response · Author response to Decision Letter 0]

5 Oct 2023

October 3, 2023

Dr. Omar Enzo Santangelo

Title: Online Search Interest in long-term symptoms of coronavirus disease 2019 During the COVID-19 Pandemic in Japan: Infodemiology Study Using the Most Visited Search Engine in Japan

Reference number: PONE-D-23-28136

Dear Dr. Omar Enzo Santangelo,

Thank you for your e-mail of September 22, 2023, regarding our manuscript, “Online Search Interest in long-term symptoms of coronavirus disease 2019 During the COVID-19 Pandemic in Japan: Infodemiology Study Using the Most Visited Search Engine in Japan”, and for the valuable comments of the reviewers. I have attached our revised manuscript, as well as a point-by-point response to the reviewers’ comments. 

We hope that the revised manuscript contains suitable responses to the comments, and we think that it has been significantly improved over the previous submission. We trust that our manuscript is now suitable for publication in PLOS ONE.

Thank you in advance for your kind consideration of our work.

Sincerely yours,

Kosuke Ishizuka, MD, PhD

Department of General Medicine, Yokohama City University School of Medicine

3-9 Fukuura, Kanazawa-ku, Yokohama-city, Kanagawa pref. Japan

Tel. +81-45-787-2800 

Fax. +81-45-350-2728

E-Mail: e103007c@yokohama-cu.ac.jp

 

RESPONSES TO REVIEWER 1:

We wish to express our appreciation to the reviewer for insightful comments that have helped us to improve our paper. We hope that the revised manuscript contains suitable responses to the comments, and we think that it has been significantly improved over the previous submission. We trust that our manuscript is now suitable for publication in PLOS ONE.

NOTE

We highlighted changes of significant issues in yellow (please see “Revised Manuscript with Track Changes”).

Comments: 

This is a paper on Online Search Interest in long-term symptoms of coronavirus disease 2019 During the COVID-19 Pandemic in Japan: Infodemiology Study Using the Most Visited Search Engine in Japan. I have some major and minor comments on your interesting and important paper. The comments are listed below.

Response: 

Thank you for your general comments. 

Per your comments and amendments, we have revised our manuscript.

In addition, we have obtained help from a professional editor to make it more understandable and free from awkward expressions. Please find a certificate of proofreading in English attached.

We answered each of the questions you posed in the revised manuscript.

Major comments:

1) Abstract: The symptoms are listed what you found in this study. I suggest that you add more numerical/statistical information on your results. You have all this information available in tables.

Response: 

Along with your suggestion, we added more numerical/statistical information on our abstract. 

In addition, we summarized the abstract due to the word limit of 300 words or less.

Changes: 

- The most frequent symptom searches with long-COVID were hair loss/baldness (3,530, 21,400, and 33,600 searches in 2020, 2021, and 2022, respectively), cough (340, 7,900 and 138,910 searches in 2020, 2021, and 2022, respectively), disturbance of smell/taste (230, 13,340, and 44,160 searches in 2020, 2021, and 2022, respectively), and headache (580, 6,180, and 42,870 searches in 2020, 2021, and 2022, respectively). (Abstract; lines 49 to 54.)

2) Introduction: Only two paragraphs introduce the topic, the first introduces long-COVID, and second briefly says that “there is a lack of data on the secular trends of internet searches using search engines.” More information (literature published) on internet searching in general is needed to add in Introduction, since this section is now too short and should be much longer in order to introduce the field of the research topic.

3) Introduction: Related to my previous comment, I suggest that you consider using a hypothesis based on the relation of long-COVID and internet searching.

Response: 

Thank you for pointing it out. Since introduction section is now too short, we added the description of introduction section in order to introduce the field of the research topic. In addition, we added more information on internet searching in general and the relation of long-COVID and internet searching.

Changes: 

- Long-COVID symptoms vary from mild, which do not interfere with daily life, to severe, which require long-term social support [2]. (Introduction; lines 74 to 75.)

- However, the study settings and follow-up methods differed from survey to survey; therefore, the results should be interpreted with caution. (Introduction; lines 78 to 80.)

- The internet is a common source of reference for information on disease and health, and internet usage influences decisions regarding initiation of care and choice of treatment in people living with long-COVID. Infodemiology (i.e., information epidemiology) is a field in health informatics defined as “the science of distribution and determinants of information in an electronic medium, specifically the internet, or in a population, with the ultimate aim to inform public health and public policy” [10]. During the COVID-19 pandemic, infodemiology studies found that online search trends for symptoms associated with COVID-19 coincided with the course of the pandemic [11]. (Introduction; lines 81 to 89.)

4) More information is needed to show where your study results can be applied to. Also, some information on future work/studies should be added in order to show which interest areas of online information seeking and/or long-COVID research should be focused on in the future. 

Response: 

As you have indicated, we added more information in order to show where our study results can be applied to. In addition, we added the description of discussion about some information on future work/studies in order to show which interest areas of online information seeking and/or long-COVID research should be focused on in the future.

This study showed a growing awareness and interest in long-COVID and documents the secular trends in the number of online searches related to long-COVID symptoms during the COVID-19 pandemic. To our knowledge, this is the first study to investigate secular trends in online interest in long-COVID. It is important to monitor interest in long-COVID on an ongoing basis to fully understand its epidemiology and the medical needs of individuals experiencing long-COVID symptoms. Additionally, a comprehensive approach is necessary for the support of individuals with long-COVID, not only for physical assessment, but also for psychological care and social reintegration support by social workers. Although this study focused on online interest in long-COVID symptoms, it studies on online interest in long-COVID treatment would also be informative.

Changes: 

- This study showed a growing awareness and interest in long-COVID and documents the secular trends in the number of online searches related to long-COVID symptoms during the COVID-19 pandemic. To our knowledge, this is the first study to investigate secular trends in online interest in long-COVID. It is important to monitor interest in long-COVID on an ongoing basis to fully understand its epidemiology and the medical needs of individuals experiencing long-COVID symptoms. Additionally, a comprehensive approach is necessary for the support of individuals with long-COVID, not only for physical assessment, but also for psychological care and social reintegration support by social workers [9, 29-31]. Although this study focused on online interest in long-COVID symptoms, it studies on online interest in long-COVID treatment would also be informative. (Discussion; lines 279 to 289.)

Minor comments:

5) The term “infodemiology” is used in this paper. Please briefly explain in text what this means.

Response: 

Along with your comments, we explained in text what the term “infodemiology” means.

Changes: 

- Infodemiology (i.e., information epidemiology) is a field in health informatics defined as “the science of distribution and determinants of information in an electronic medium, specifically the internet, or in a population, with the ultimate aim to inform public health and public policy” [10]. (Introduction; lines 83 to 86.)

6) “Internet” and “internet” are used in text, sometimes with a capital letter, sometimes without. Please choose one of them and use consistently throughout.

Response: 

Thank you for your comments. We chose one of them and use consistently throughout.

Changes: 

- The internet user population was defined as the number of internet users throughout Japan, calculated based on the “Telecommunications Usage Trends Survey,” published by the Japan Ministry of Internal Affairs and Communications. (Methods; lines 111 to 113.)

- In this study, we conducted infodemiology using data on internet searches using the Yahoo search engine. (Methods; lines 132 to 133.)

- Between 2020 and 2022, 50–51% of all adult internet users were male, and the user population was relatively evenly distributed across different age groups (Table 1). (Results; lines 159 to 160.)

- a The internet user population is the number of internet users throughout Japan, calculated based on the "Telecommunications Usage Trends Survey" published by the Ministry of Internal Affairs and Communications. (Results; lines 171 to 173.)

- First, the data source was limited to Yahoo! JAPAN and does not include internet searches conducted using other search engines. (Limitations; lines 292 to 293.)

- Fifth, the number of internet users and the number of users of the Yahoo! JAPAN search engine did not increase between 2020 and 2022, which may not be related to the trend in Japan. (Limitations; lines 300 to 302.)

- Sixth, internet users may search for long-COVID symptoms when prolonged symptoms occur after COVID-19, even if the symptoms do not persist for more than 2 months, which is the definition of long-COVID. (Limitations; lines 302 to 305.)

7) References: Please use PMIDs or DOIs at the end of each scientific reference.

Response: 

Along with your recommendation, we used DOIs at the end of each scientific reference.

Changes: 

- References (References; lines 331 to 431.)

Others

We also indicated other minor corrections in yellow markers (please see “Revised Manuscript with Track Changes”).

We thank the reviewer for such pertinent comments. We hope that the revised manuscript contains suitable responses to the comments, and we think that it has been significantly improved compared to the previous submission. We trust that our manuscript is now suitable for publication in PLOS ONE.

 

RESPONSES TO REVIEWER 2:

We wish to express our appreciation to the reviewer for insightful comments that have helped us to improve our paper. We hope that the revised manuscript contains suitable responses to the comments, and we think that it has been significantly improved over the previous submission. We trust that our manuscript is now suitable for publication in PLOS ONE.

NOTE

We highlighted changes of significant issues in yellow (please see “Revised Manuscript with Track Changes”).

Comments: 

The authors perform an internet search analysis related to Long COVID and symptoms. They use Yahoo analytics, analyze the dataset retrieved, and interpret the findings.

Response: 

Thank you for your general comments. 

Per your comments and amendments, we have revised our manuscript.

In addition, we have obtained help from a professional editor to make it more understandable and free from awkward expressions. Please find a certificate of proofreading in English attached.

We answered each of the questions you posed in the revised manuscript.

1) Table 2: commas are misplaced in the numbers for the top row.

Response: 

As you have indicated, we revised comma in the numbers for the top row in Table 2.

Changes: 

- Table 2. (Results; line 198.)

2) Page 11, line 90: Link to the manual doesn't work.

Response: 

As you pointed it out, we reconfirmed that link to the manual works.

Since that link is written in Japanese, we added the additional explanation of Yahoo! Japan DS. INSIGHT functions in English in S1 Table.

Changes: 

- The user manual can be found at https://datasolution.yahoo-net.jp/view/knowledgebase/641. (Methods; lines 109 to 110.)

- S1 Table. 

3) Data accessibility: The dataset should be provided directly, or clearer process for how readers might directly obtain the data should be provided. The links to the Japanese language websites were unclear as to how the data might be downloaded for independent analysis.

Response: 

Along with your recommendation, we added clearer process for how readers might directly obtain the data should be provided. Yahoo! Japan DS. INSIGHT can be accessed by people working at educational and research institutions by obtaining a login ID. It is a tool that enables analysis of all big behavioral data of Yahoo! Japan, such as keywords, time period, gender, age, and prefecture, and the transitions and trends in these searches over time. We added the explanation about the system of the Yahoo! Japan DS. INSIGHT.

In addition, the linked dataset is written in Japanese, not English. We have added an explanation of Yahoo! Japan DS. INSIGHT functions in English in the main text and S1 Table for readers who cannot read Japanese. We added about this point in the Limitations section.

Changes: 

- Yahoo! Japan DS. INSIGHT can be accessed by people working at educational and research institutions by obtaining a login ID. It is a tool that enables analysis of all big behavioral data of Yahoo! Japan, such as keywords, time period, gender, age, and prefecture, and the transitions and trends in these searches over time (S1 Table). (Methods; lines 105 to 109.)

- Eighth, the linked dataset is written in Japanese, not English. We have added an explanation of Yahoo! Japan DS. INSIGHT functions in English in the main text and S1 Table for readers who cannot read Japanese. (Limitations; lines 308 to 311.)

4) Comparison/adjudication of the evaluations performed by the independent researchers to minimize bias does not appear to be reported. What was the process followed and how often were there differences?

Response: 

Thank you for your comments. We compared the symptoms searched with long-COVID and analyzed the difference in online interest in each long-COVID symptom from January 2020 to December 2022, by year. Long-COVID symptoms were classified according to “Component 1—Symptoms and complaints” of the International Classification of Primary Care, 2nd edition (ICPC-2) [14]. The ICPC focuses on an analysis of individuals’ “reason for the encounter,” and it has the advantage that it can directly describe the reason for the encounter from a patient perspective as well as from a healthcare provider perspective. In this study, we conducted infodemiology using data on internet searches using the Yahoo search engine. To be sure, two researchers (KI and TM) conducted independent evaluations of the Yahoo search engine data for symptom classification in order to minimize observer bias. Classification of the symptoms searched with long-COVID into ICPC-2 terms by two independent researchers (KI and TM) was in complete agreement. We added our manuscript about this point.

Changes: 

- Classification of the symptoms searched with long-COVID into ICPC-2 terms by two independent researchers (KI and TM) was in complete agreement. (Results; lines 180 to 181.)

5) Data by prefecture is noted as retrieved yet not reported. This may be of interest to readers including per capita annual searches by prefecture and implications.

Response: 

As you have indicated, we considered that data by prefecture is of interest to readers including per capita annual searches by prefecture and implications. Along with your recommendation, we added the data on the number of annual searches per capita by prefecture in supplemental file.

Changes: 

- Additionally, the data on the number of annual searches per capita by prefecture are shown in S2 Table. (Results; lines 166 to 167.)

- S2 Table.

6) Stratification by age and sex are reported yet the reliability of this information is not discussed. How does Yahoo! have access to this data? If this merely represents the Yahoo! user login info, this may be inaccurate, not reflect the current user, and, aren't many searches anonymous? For example, is there published data for how often other users in a family access Yahoo! that's already logged in under another family member? The method by which user age/sex are inferred and these limitations should be discussed fully along with any references that validate usage of these data elements in this manner.

Response: 

Thank you for your comments. Stratification by age and gender represents the Yahoo! user login info, but may not reflect the current user because Yahoo! IDs can be used by multiple users with different demographics. For example, users in a family may access Yahoo! using the Yahoo! ID of another family member. We added about this point in the limitations section.

Changes: 

- Seventh, stratification by age and gender represents the Yahoo! user login info, but may not reflect the current user because Yahoo! IDs can be used by multiple users with different demographics. For example, users in a family may access Yahoo! using the Yahoo! ID of another family member. (Limitations; lines 305 to 308.)

7) Did the authors consider obtaining baseline search data for each of these symptoms pre-pandemic and also during the pandemic as distinct searches, not paired with Long Covid? This would clarify whether these symptom findings represent an absolute increase in searches for these symptoms or simply shifted time-wise when COVID was prevalent. In addition, the isolated symptom trend if including prior years would clarify whether a steady baseline existed pre-pandemic for these isolated search terms or whether, for some reason, there was already a positive total trend that's just clumping around COVID peaks.

Response: 

Along with your recommendation, we added baseline data on searches for each of these symptoms during the pandemic as distinct searches, not paired with long-COVID in Table 3. 

Table 3 clarified that a steady baseline during the pandemic for these isolated search terms not paired with long-COVID. We added the discussion about this point.

However, because search data was only available after 2020, when the pandemic began, we could not determine whether the number of searches represent an absolute increase in the number of searches for these symptoms or simply fluctuations in the number of searches over time during the period when COVID-19 was prevalent. We added about this point in the limitations section.

Changes: 

- In addition, we obtained baseline data on searches for each of these symptoms during the pandemic as distinct searches, not paired with long-COVID. (Methods; lines 130 to 132.)

- In addition, Table 3 shows the search data for each symptom included in Table 2 as distinct searches, not paired with long-COVID. (Results; lines 194 to 196.)

- Table 3.

- Furthermore, Table 3 clarified that a steady baseline during the pandemic for these isolated search terms not paired with long-COVID. (Discussion; lines 216 to 218.)

- Nineth, because search data was only available after 2020, when the pandemic began, we could not determine whether the number of searches represent an absolute increase in the number of searches for these symptoms or simply fluctuations in the number of searches over time during the period when COVID-19 was prevalent. (Limitations; lines 311 to 315.)

8) Since Long COVID symptoms by definition would appear months after COVID infection, the authors should address why peaks in Long COVID searches appear to coincide almost exactly with the number of active cases and are not delayed.

Response: 

Thank you for pointing it out. By definition, long-COVID symptoms appear months after COVID-19. Therefore, although internet users were worried about long-COVID, these findings may represent persistent acute symptoms present before the time threshold for long-COVID onset. We added the discussion about this point.

Changes: 

- By definition, long-COVID symptoms appear months after COVID-19. Therefore, although internet users were worried about long-COVID, these findings may represent persistent acute symptoms present before the time threshold for long-COVID onset. (Discussion; lines 208 to 211.)

Overall - nice analysis. Including baselines for the symptoms analyzed would make the paper stronger and more compellingly linked to COVID than search epiphenomena clustered during outbreaks. Absence of a time shift in the search also introduces the notion that these findings represented acute symptoms persisting prior to the the Long COVID time threshold although patients were worried about long COVID. One other point that may be of interest would be putative COVID (and maybe Long COVID) treatments paired to these search terms and trends over time.

Response: 

Thank you for your comments. 

Along with your recommendation, we obtained baseline data on searches for each of these symptoms during the pandemic as distinct searches, not paired with long-COVID in Table 3. 

Table 3 clarified that a steady baseline during the pandemic for these isolated search terms not paired with long-COVID. We added the discussion about this point.

However, because search data was only available after 2020, when the pandemic began, we could not determine whether the number of searches represent an absolute increase in the number of searches for these symptoms or simply fluctuations in the number of searches over time during the period when COVID-19 was prevalent. We added about this point in the limitations section.

By definition, long-COVID symptoms appear months after COVID-19. Therefore, although internet users were worried about long-COVID, these findings may represent persistent acute symptoms present before the time threshold for long-COVID onset. We added the discussion about this point.

In addition, we added the description of discussion about some information on future work/studies in order to show which interest areas of online information seeking and/or long-COVID research should be focused on in the future. This study showed a growing awareness and interest in long-COVID and documents the secular trends in the number of online searches related to long-COVID symptoms during the COVID-19 pandemic. To our knowledge, this is the first study to investigate secular trends in online interest in long-COVID. It is important to monitor interest in long-COVID on an ongoing basis to fully understand its epidemiology and the medical needs of individuals experiencing long-COVID symptoms. Additionally, a comprehensive approach is necessary for the support of individuals with long-COVID, not only for physical assessment, but also for psychological care and social reintegration support by social workers [9, 29-31]. Although this study focused on online interest in long-COVID symptoms, it studies on online interest in long-COVID treatment would also be informative.

Changes: 

- In addition, we obtained baseline data on searches for each of these symptoms during the pandemic as distinct searches, not paired with long-COVID. (Methods; lines 130 to 132.)

- In addition, Table 3 shows the search data for each symptom included in Table 2 as distinct searches, not paired with long-COVID. (Results; lines 194 to 196.)

- By definition, long-COVID symptoms appear months after COVID-19. Therefore, although internet users were worried about long-COVID, these findings may represent persistent acute symptoms present before the time threshold for long-COVID onset. (Discussion; lines 208 to 211.)

- Furthermore, Table 3 clarified that a steady baseline during the pandemic for these isolated search terms not paired with long-COVID. (Discussion; lines 216 to 218.)

- This study showed a growing awareness and interest in long-COVID and documents the secular trends in the number of online searches related to long-COVID symptoms during the COVID-19 pandemic. To our knowledge, this is the first study to investigate secular trends in online interest in long-COVID. It is important to monitor interest in long-COVID on an ongoing basis to fully understand its epidemiology and the medical needs of individuals experiencing long-COVID symptoms. Additionally, a comprehensive approach is necessary for the support of individuals with long-COVID, not only for physical assessment, but also for psychological care and social reintegration support by social workers [9, 29-31]. Although this study focused on online interest in long-COVID symptoms, it studies on online interest in long-COVID treatment would also be informative. (Discussion; lines 279 to 289.)

- Nineth, because search data was only available after 2020, when the pandemic began, we could not determine whether the number of searches represent an absolute increase in the number of searches for these symptoms or simply fluctuations in the number of searches over time during the period when COVID-19 was prevalent. (Limitations; lines 311 to 315.)

Others

We also indicated other minor corrections in yellow markers (please see “Revised Manuscript with Track Changes”).

We thank the reviewer for such pertinent comments. We hope that the revised manuscript contains suitable responses to the comments and has been significantly improved compared to the previous submission. We trust that our manuscript is now suitable for publication in PLOS ONE.

---

## [Decision Letter · Decision Letter 1]

12 Oct 2023

PONE-D-23-28136R1Online Search Interest in long-term symptoms of coronavirus disease 2019 During the COVID-19 Pandemic in Japan: Infodemiology Study Using the Most Visited Search Engine in JapanPLOS ONE

Dear Dr. Ishizuka,

Thank you for submitting your manuscript to PLOS ONE. After careful consideration, we feel that it has merit but does not fully meet PLOS ONE’s publication criteria as it currently stands. Therefore, we invite you to submit a revised version of the manuscript that addresses the points raised during the review process.

ACADEMIC EDITOR: Minor revision

We look forward to receiving your revised manuscript.

Kind regards,

Omar Enzo Santangelo

Academic Editor

PLOS ONE

Journal Requirements:

Reviewers' comments:

Reviewer's Responses to Questions

**Comments to the Author**

1. If the authors have adequately addressed your comments raised in a previous round of review and you feel that this manuscript is now acceptable for publication, you may indicate that here to bypass the “Comments to the Author” section, enter your conflict of interest statement in the “Confidential to Editor” section, and submit your "Accept" recommendation.

Reviewer #1: All comments have been addressed

Reviewer #2: All comments have been addressed

2. Is the manuscript technically sound, and do the data support the conclusions?

Reviewer #1: Yes

Reviewer #2: Yes

3. Has the statistical analysis been performed appropriately and rigorously? 

Reviewer #1: Yes

Reviewer #2: I Don't Know

4. Have the authors made all data underlying the findings in their manuscript fully available?

Reviewer #1: Yes

Reviewer #2: Yes

5. Is the manuscript presented in an intelligible fashion and written in standard English?

Reviewer #1: Yes

Reviewer #2: Yes

6. Review Comments to the Author

Reviewer #1: I read your revised paper throughout, and it has definitely improved. I still have one minor comment:

1. Reference numbers 3 and 14: Are these books or what kind of references? Please add online links if available.

Reviewer #2: Nice job incorporating reviewers' comments. Two items:

1. "nineth" is misspelled

2. Consider including when Long-Covid was first reported. This is helpful to know for any temporal analysis of this term.

7. PLOS authors have the option to publish the peer review history of their article (what does this mean?). If published, this will include your full peer review and any attached files.

Reviewer #1: No

Reviewer #2: **Yes: **David Liebovitz

---

## [Author Response · Author response to Decision Letter 1]

27 Oct 2023

October 27, 2023

Dr. Omar Enzo Santangelo

Title: Online Search Interest in long-term symptoms of coronavirus disease 2019 During the COVID-19 Pandemic in Japan: Infodemiology Study Using the Most Visited Search Engine in Japan

Reference number: PONE-D-23-28136.R1

Dear Dr. Omar Enzo Santangelo,

Thank you for your e-mail of October 13, 2023, regarding our manuscript, “Online Search Interest in long-term symptoms of coronavirus disease 2019 During the COVID-19 Pandemic in Japan: Infodemiology Study Using the Most Visited Search Engine in Japan”, and for the valuable comments of the reviewers. I have attached our revised manuscript, as well as a point-by-point response to the reviewers’ comments. 

We hope that the revised manuscript contains suitable responses to the comments, and we think that it has been significantly improved over the previous submission. We trust that our manuscript is now suitable for publication in PLOS ONE.

Thank you in advance for your kind consideration of our work.

Sincerely yours,

Kosuke Ishizuka, MD, PhD

Department of General Medicine, Yokohama City University School of Medicine

3-9 Fukuura, Kanazawa-ku, Yokohama-city, Kanagawa pref. Japan

Tel. +81-45-787-2800 

Fax. +81-45-350-2728

E-Mail: e103007c@yokohama-cu.ac.jp

 

RESPONSES TO REVIEWER 1:

We wish to express our appreciation to the reviewer for insightful comments that have helped us to improve our paper. We hope that the revised manuscript contains suitable responses to the comments, and we think that it has been significantly improved over the previous submission. We trust that our manuscript is now suitable for publication in PLOS ONE.

NOTE

We highlighted changes of significant issues in yellow (please see “Revised Manuscript with Track Changes”).

Comments: 

I read your revised paper throughout, and it has definitely improved. I still have one minor comment:

Response: 

Thank you for your general comments. 

Per your comments and amendments, we have revised our manuscript.

1) Reference numbers 3 and 14: Are these books or what kind of references? Please add online links if available.

Response: 

Thank you for pointing it out.

We added the References number 3-6, so Reference number 3 in the previous manuscript corresponds to Reference number 7 in the revised manuscript, and Reference number 14 in the previous manuscript corresponds to Reference number 18 in the revised manuscript.

Reference number 7 in the revised manuscript is an online article. We added the online link.

In addition, Reference number 18 in the revised manuscript is a book as following link; https://books.google.co.jp/books/about/ICPC_2.html?id=Wr_3OqVxypMC&redir_esc=y. We reconfirmed that the reference is correctly described according to the PLOS ONE submission guidelines.

Changes: 

- References (References; lines 340 to 454.)

Others

We also indicated other minor corrections in yellow markers (please see “Revised Manuscript with Track Changes”).

We thank the reviewer for such pertinent comments. We hope that the revised manuscript contains suitable responses to the comments, and we think that it has been significantly improved compared to the previous submission. We trust that our manuscript is now suitable for publication in PLOS ONE.

 

RESPONSES TO REVIEWER 2:

We wish to express our appreciation to the reviewer for insightful comments that have helped us to improve our paper. We hope that the revised manuscript contains suitable responses to the comments, and we think that it has been significantly improved over the previous submission. We trust that our manuscript is now suitable for publication in PLOS ONE.

NOTE

We highlighted changes of significant issues in yellow (please see “Revised Manuscript with Track Changes”).

Comments: 

Nice job incorporating reviewers' comments. Two items:

Response: 

Thank you for your general comments. 

Per your comments and amendments, we have revised our manuscript.

In addition, we have obtained help from a professional editor to make it more understandable, and free from awkward expressions. Please find a certificate of proofreading in English attached.

We answered each of the questions you posed in the revised manuscript.

1) "nineth" is misspelled.

Response: 

Thank you for pointing it out. Along with your comments, we revised the description.

Changes: 

- Ninth, because search data was only available after 2020, when the pandemic began, we could not determine whether the number of searches represent an absolute increase in the number of searches for these symptoms or simply fluctuations in the number of searches over time during the period when COVID-19 was prevalent. (Limitations; lines 320 to 324.)

2) Consider including when Long-Covid was first reported. This is helpful to know for any temporal analysis of this term.

Response: 

As you have indicated, we included when long-COVID was first reported as follows:

Changes: 

- The term “long-COVID” first entered public discourse on May 20, 2020, when Dr. Elisa Perego shared #longcovid on Twitter, now known as X [3-5]. A hashtag (#) is a concept-labelling tool that encourages public sharing and worldwide dissemination of the discussed concept by grouping conversations around it. Initially, #longcovid referred to Perego's experience of prolonged “cyclical, multiphasic, and multisystem” symptoms of COVID-19 [5]. Twitter, with approximately 187 million users worldwide at the time, emerged as a platform for individuals with persistent COVID-19 symptoms to publicly share their experiences, symptoms, and concerns associated with living with the condition [5, 6]. Subsequently, on October 6, 2021, the World Health Organization (WHO) defined long-COVID as the onset of symptoms within 3 months after the onset of COVID-19 and persistence of symptoms for more than 2 months, excluding other diseases [7, 8]. (Introduction; lines 70 to 81.)

Others

We also indicated other minor corrections in yellow markers (please see “Revised Manuscript with Track Changes”).

We thank the reviewer for such pertinent comments. We hope that the revised manuscript contains suitable responses to the comments and has been significantly improved compared to the previous submission. We trust that our manuscript is now suitable for publication in PLOS ONE.

---

## [Editor Report · Decision Letter 2]

30 Oct 2023

Online Search Interest in long-term symptoms of coronavirus disease 2019 During the COVID-19 Pandemic in Japan: Infodemiology Study Using the Most Visited Search Engine in Japan

PONE-D-23-28136R2

Dear Dr. Ishizuka,

We’re pleased to inform you that your manuscript has been judged scientifically suitable for publication and will be formally accepted for publication once it meets all outstanding technical requirements.

Kind regards,

Omar Enzo Santangelo

Academic Editor

PLOS ONE
---

## [Editor Report · Acceptance letter]

6 Nov 2023

PONE-D-23-28136R2 

Online Search Interest in long-term symptoms of coronavirus disease 2019 During the COVID-19 Pandemic in Japan: Infodemiology Study Using the Most Visited Search Engine in Japan 

Dear Dr. Ishizuka:

I'm pleased to inform you that your manuscript has been deemed suitable for publication in PLOS ONE. Congratulations! Your manuscript is now with our production department. 

Kind regards, 

on behalf of

Dr. Omar Enzo Santangelo 

Academic Editor

PLOS ONE